# Towards Autonomous Agents: Adaptive-planning, Reasoning, and Acting in Language Models

## Abstract

We propose a novel in-context learning algorithm for building autonomous decision-making language agents. The language agent continuously attempts to solve the same task by reasoning, acting, observing and then self-correcting each time the task fails. Our selected language agent demonstrates the ability to solve tasks in a text-based game environment. Our results show that the gemma-2-9b-it language model, using our proposed method, can successfully complete two of six tasks that failed in the first attempt. This highlights the effectiveness of our approach in enhancing the problem-solving capabilities of a single language model through self-correction, paving the way for more advanced autonomous agents. The code is publicly available at `https://anonymous.4open.science/r/AutonomousLLMAgentwithAdaptingPlanning-D613/`.

## 1 Introduction

Large language models (LLMs) are large statistical models that predict the next word, phrase, sentence, or paragraph based on a given input (Demszky et al., 2023). The quality of the output from a language model can be heavily influenced by the input prompt it receives (Arvidsson & Axell, 2023). One of the capabilities of LLMs is in-context learning, where they learn a new task from a small set of exemplars provided in the prompt during inference (Minaee et al., 2024). Prompt engineering is the process of designing and refining input prompts to elicit desired responses from LLMs (Ekin, 2023).

In Chain-of-Thought (CoT) (Wei et al., 2022), given a prompt with exemplars that include an input part and an output part, a chain of thought consists of a series of intermediate natural language reasoning steps added between the input and output parts in each exemplar to produce the final output. However, the CoT prompting doesn't have the ability to update its knowledge from the external world. ReAct (Yao et al., 2023) prompting addresses the problem by providing the language model with a prior language description to guide its reasoning about solving diverse language reasoning and decision making tasks and adapting this reasoning by acting on and receiving the feedback from the external world. In Reflexion (Shinn et al., 2023), they proposed autonomous decision-making LLM agents by adding a reflection step to the CoT or ReAct prompt to adjust the reasoning, facilitating language agents' learning from prior failings through verbal reinforcement. VOYAGER (Wang et al., 2023) is a LLM based agent designed to explore an open-ended world and attain diverse skills through the integration of automatic curriculum, skill library management, and an iterative prompting mechanism incorporating environmental feedback, execution errors, and self-verification to enhance program performance. In Motif (Klissarov et al., 2024), an agent is trained through reinforcement learning to maximize rewards from a parameterized model, which is trained based on preferences selected by a language model over pairs of actions, aimed at achieving a specific goal in a given environment.

In this study, we propose Self-Adaptive Language Agent (SALA), which is an adaptive decision-making language agent that self-adjusts the reasoning process of ReAct with a correction mechanism from Reflexion. Here, we let the language model agent adapt its own policy by correcting its previous failure using its internal knowledge. The proposed SALA differs from Reflexion (Shinn et al., 2023), which uses two LLMs: one for action generation and another for reflection whereas SALA

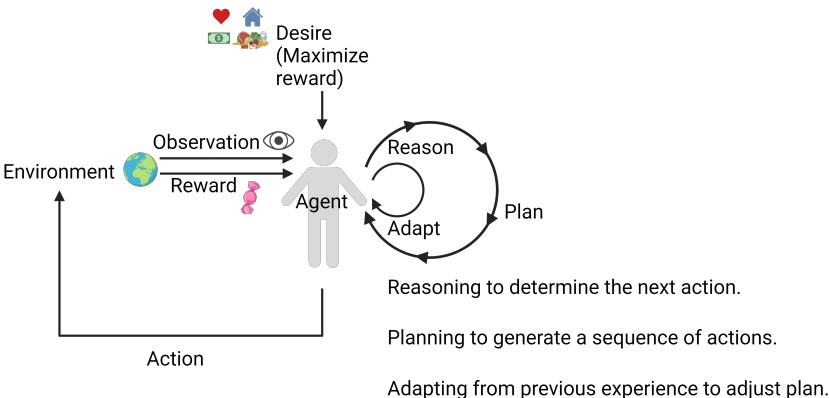

Figure 1: An architecture towards autonomous agent. Created with BioRender.com.

employs a single LLM that can self-adapt its reasoning and acting behavior making it an autonomous language agent. Our experimental results show that in twelve different decision making tasks from the ALFWorld (Shridhar et al., 2020) environment, the proposed SALA achieves a success rate of approximately 83%, which is higher than the ReAct-based agent, which achieved a success rate of 67%. In addition, the SALA could solve two tasks that couldn't be completed in the first trail, demonstrating the effectiveness of our approach.

## 2 METHODS

Let a text game be denoted as a function $f$ that maps the state $s \in \mathbb{V}$ and the action $a \in \mathbb{V}$ to an observation $o \in \mathbb{V}$, where $\mathbb{V}$ is a set of vocabulary. Let $\pi_\theta$ be an LLM agent over a pre-trained set of parameters $\theta$. Let $s_0$ be the initial state of the environment $f$, we aim to produce a sequence of actions $(a_0, a_1, a_2, \dots)$, where $a_i \in \mathbb{V}$ for $i \in \mathbb{Z}$, to change the state to a terminal state that indicates the game is cleared.

In ReAct prompting (Yao et al., 2023), they propose to use an LLM to produce the thought $(t_0, t_1, t_2, \dots)$, where $t_i \in \mathbb{V}$ for $i \in \mathbb{Z}$, by $t_i \sim \pi_\theta(t_i|s_i)$, where $s_i = \{s_{i-1}, t_{i-1}, a_{i-1}, o_i\}$ for $i \in \mathbb{Z}^+$, $a_i \sim \pi_\theta(a_i|s_i, t_i)$ for $i \in \mathbb{Z}$, and $o_{i+1} = f(s_i, a_i)$ for $i \in \mathbb{Z}$.

The limitation of ReAct prompting (Yao et al., 2023) is that complex tasks with a large action space require more demonstrations to learn effectively. The LLM may produce incorrect actions that do not lead to task completion. Reflexion (Shinn et al., 2023) addresses this problem by using an additional LLM to iteratively provide reflection text that will be added to the ReAct prompt for improvement. More specifically, for each trial $ep \in \mathbb{Z}^+$, if $ep \geq N$, where $N$ is a maximum number for each trail, a self-reflection $r^{ep}$ is generated and a new state $s_0^{ep+1} = \{s_N^{ep+1}, r^{ep}\}$ is formed to be used as the initial state in the next trial $ep + 1$. However, the method in Reflexion (Shinn et al., 2023) necessitates two LLMs, where one LLM is used to generate the thought or action, and another LLM is used to generate the reflection. We will modify this by using a single LLM to generate thought, action, and self-adaptation, which is the correction from the previous failed trial.

The architecture of the main idea of our work is shown in Figure 1. A desire is provided to an agent to motivate it to solve a specific task in a given environment. The agent can perform an action to interact with the environment, causing the state of the environment to change. The agent then receives an observation that describes the status of the environment and a reward signal. The action may be proposed from three different processes: the reasoning process determines the next action based on the current progress; the planning process proposes a series of actions that can be used to solve a specific task; and the adaptation process summarizes previous progress to provide a better plan towards maximizing the reward.

We present a novel algorithm in Algorithm 1. Initially, we have the initial state $s_0$ which provides instructions, presents exemplars, and describes the environment and the goal for a specific task. $\pi_\theta$ is an LLM agent with a set of parameters $\theta$. $\tau = \{s_0, t_0, a_0, o_1, \dots\}$ is a sequence of the concatenation of state, thought, action, and observation, where $s_k$, $t_k$, $a_k$, and $o_k$ are sequences of tokens representing the $k$-th state, thought, action, and observation for $k \in \mathbb{Z}$, respectively. The return $R(\tau)$ is a string indicating whether the task is completed or not. $ep$ is a variable indicating the number of trials. The environment is reinitialized at each trial.

At the first time step $k = 0$, the thought is sampled from

$$t_0^1 \sim \pi_\theta(t_0^1|s_0^1), \tag{1}$$

where the subscript $0$ indicates the first time step, and the superscript $1$ indicates the first trail. $t_0^1$ represents the first thought in the first trial, and $s_0^1$ represents the first state in the first trial, with both being sequences of tokens. The action is then sampled from

$$a_0^1 \sim \pi_\theta(a_0^1|s_0^1, t_0^1), \tag{2}$$

where $a_0^1$ represents the first action in the first trial, and the action is a sequence of tokens. The second observation in the first trial, $o_1^1$, is a sequence of tokens obtained by executing the action $a_0^1$ in the environment $f$ at state $s_0^1$ as

$$o_1^1 = f(s_0^1, a_0^1). \tag{3}$$

A new state $s_1^1$ is formed by concatenating the thought $t_0^1$, action $a_0^1$, and observation $o_1^1$ after state $s_0^1$ as

$$s_1^1 = \{s_0^1, t_0^1, a_0^1, o_1^1\}. \tag{4}$$

If a maximum time step is reached, the task fails and the return $R(\tau)$ is concatenated with "New plan: ". They are concatenated after the current state of the environment $s_{50}^1$ to form the initial state in the next trial $s_0^2$ as

$$s_0^2 = \{s_{50}^1, R(\tau)\}, \tag{5}$$

where $\tau = \{s_0, t_0, a_0, o_1, t_1, a_1, o_2, t_2, \dots o_{50}\}$. In the next trial, the first thought in the second trial, $t_0^2$, is sampled from the LLM by

$$t_0^2 \sim \pi_\theta(t_0^2|s_0^2), \tag{6}$$

We call the initial thought $t_0^{ep}$ at the $ep$-th trial for $ep > 1$ as the adaptation from the $(ep-1)$-th trail and $t_0^{ep}$ indicates the correction of the $(ep-1)$-th failed trail to improve the next trail. In the next step, we propose to replace the initial state in the second trial with the initial state in the first trail to reduce the context length. We call this step compression. By performing compression, the first action in the second trail will only be conditioned on the initial state in the first trail $s_0^1$ and the adaptation from the first trail $t_0^2$ as

$$a_0^2 \sim \pi_\theta(a_0^2|s_0^2, t_0^2). \tag{7}$$

---

**Algorithm 1** Self-Adaptive Language Agent

Initialize the world state $s_0$ as a text of exemplars and task, where each token $\in Vocab$.
Let $\pi_\theta$ be a LLM agent over a pre-trained set of parameters $\theta$.
Let a trajectory $\tau = \{s_0, t_0, a_0, o_1, \dots\}$ be a sequence of state, thought, action, and observation.
Let $R(\tau)$ be the return for trajectory $\tau$.
Let $ep = 1$.
While $R(\tau) \neq "OK"$ do
    Let $k = 0$.
    While $k < 50 \ || \ R(\tau) = "OK"$ do
        Generate thought $t_k^{ep} \sim \pi_\theta(t_k^{ep}|s_k^{ep})$.
        Compression step:
            If $k = 0$, then $s_0^{ep} = s_0$.
        Generate action $a_k^{ep} \sim \pi_\theta(a_k^{ep}|s_k^{ep}, t_k^{ep})$.
        Get observation $o_{k+1}^{ep} = f(s_k^{ep}, a_k^{ep})$.
        Let $s_{k+1}^{ep} = \{s_k^{ep}, t_k^{ep}, a_k^{ep}, o_{k+1}^{ep}\}$.
        $k := k + 1$
    Concatenate $R(\tau)$ with "New plan: ".
    $s_0^{ep+1} = \{s_k^{ep}, R(\tau)\}$
    $ep := ep + 1$

---

## 3    THE ALFWORLD ENVIRONMENT

There are six types of tasks in the ALFWorld environment (Shridhar et al., 2020): Pick and Place, Examine in Light, Clean and Place, Heat and Place, Cool and Place, and Pick Two and Place. For each task, a description of available receptacles is given in the first part of the instruction as follows: You are in the middle of a room. Looking quickly around you, you see a {recep1 id}, a {recep2 id}, . . . , and a {recepN id}, where recepN refers to the Nth receptacles like drawers or cabinet and id$\in \mathbb{Z}^+$. An example is shown in the text in Figure 2 with a green background.

After the description of the available receptacles, the goal instructions are provided based on the six different task types. For a Pick and Place task, the instruction will be either "put a obj in recap" or "put some obj on recap". For an Examine in Light task, it will be either "look at obj under the lamp" or "examine the obj with the lamp". For a Clean and Place task, it will be either "put a clean obj in recap" or "clean some obj and put it in recap". For a Heat and Place task, it will be either "put a hot obj in recap" or "heat some obj and put it in recap". For a Cool and Place task, it will be either "put a cool obj in recap" or "cool some obj and put it in recap". For a Pick Two and Place task, it will be either "put two obj in recap" or "find two obj and put them in recap". Inside the curly brackets, {obj}, {recep} and {lamp} refer to object, receptacle, and lamp classes, respectively. An example is shown in the text in Figure 2 with a red background.

After the goal instruction of the task, the agent or the user can interact with the game environment using the following nine different text actions: go to recap id, open recap id, clean obj id with recap id, take obj id from recap id, close recap id, close recap id, heat obj id with recap id, put obj id in/on recap id, and use recap id. Given the action, the game environment will return text observations accordingly. An example is shown in the text in Figure 2 with a magenta background.

## 4    EXPERIMENTAL DESIGNS AND RESULTS

### 4.1    EXPERIMENT ON REACT PROMPTING IN THE ALFWORLD ENVIRONMENT

In ReAct (Yao et al., 2023), they randomly annotate the trajectories for each task type. Each trajectory includes sparse thoughts that decompose the goal, track subgoal completion, determine the next subgoal, and use commonsense reasoning to find an object and determine what to do with it. An exemplar used in ReAct (Yao et al., 2023) is shown in Figure 3. The instruction text, "Interact with a household to solve a task. Here are two examples.", is added above the exemplars to indicate that the general goal is to complete a household task, and the texts below it include two exemplars. Following the examples, the text 'Here is the task.' is concatenated, followed by the description of available receptacles and the description of the goal instruction from the ALFWorld environment (Shridhar et al., 2020). Finally, the greater-than symbol ('>') is added to indicate that the language model should generate text after it, starting with actions or thoughts.

ReAct (Yao et al., 2023) evaluated their method on 134 tasks in the ALFWorld environment (Shridhar et al., 2020), achieving a success rate of 70.9% using PaLM-540B (Chowdhery et al., 2023) and 78.4% using the GPT-3 text-davinci-002 model (Brown et al., 2020), where each output token is selected with the highest probability. However, PaLM-540B (Chowdhery et al., 2023) is not publicly available, and the text-davinci-002 model (Brown et al., 2020) was shut down by OpenAI on January 4th, 2024. Due to the difficulty in reproducing the results from (Yao et al., 2023), we tested various open-source LLMs and selected the one with the highest success rate to develop our method. The results in Table 1 show that the gemma-2-9b-it model outperforms other models with a success rate of 62% in a 12-hour run. We will use gemma-2-9b-it for the subsequent experiments.

We observed that gemma-2-9b-it model outperforms other models (gemma-2-9b, Mistral-7B-v0.3, Mistral-7B-Instruct-v0.3, Llama-2-7b-hf, Phi-3-medium-128k-instruct, deepseek-llm-7b-base, and zephyr-7b-alpha) with a success rate of 40% in solving various tasks in the ALFWorld environment (Shridhar et al., 2020) using ReAct prompting (Yao et al., 2023). We identified three common issues with these LLMs. First, the LLMs may attempt to retrieve an object from a location where the object does not exist, repeatedly performing the same action until reaching the maximum number of steps, 50. Second, the LLMs may select an incorrect item. For instance, when the task is to "put a clean cloth in countertop", the gemma-2-9b model may pick up a "handtowel" instead. Subsequently, the LLM cannot complete the task by cleaning the "handtowel", resulting in repeated attempts to clean

```
You are in the middle of a room. Looking quickly around you,
you see a cabinet 6, a cabinet 5, a cabinet 4, a cabinet 3, a
cabinet 2, a cabinet 1, a coffeemachine 1, a countertop 3, a
countertop 2, a countertop 1, a drawer 3, a drawer 2, a
drawer 1, a fridge 1, a garbagecan 1, a microwave 1, a shelf
3, a shelf 2, a shelf 1, a sinkbasin 1, a stoveburner 4, a
stoveburner 3, a stoveburner 2, a stoveburner 1, and a
toaster 1.
```

**(1) A description of available receptacles**

```
Your task is to: cool some pan and put it in countertop.
```

**(2) A description of the goal instruction**

```
> go to stoveburner 1
On the stoveburner 1, you see a pot 1.
> go to stoveburner 2
On the stoveburner 2, you see a pan 1.
> take pan 1 from stoveburner 2
You pick up the pan 1 from the stoveburner 2.
> go to fridge 1
The fridge 1 is closed..
> cool pan 1 with fridge 1
You cool the pan 1 using the fridge 1.
> go to countertop 1
On the countertop 1, you see a mug 1.
> put pan 1 in/on countertop 1
You put the pan 1 in/on the countertop 1.
```

**(3) A sequence of actions and observations.**

Figure 2: A trajectory in the ALFWorld environment (Shridhar et al., 2020). The text in the black box is composed of the description of available receptacles, the description of the goal instruction, and a sequence of actions and observations. (1) The description of available receptacles is listed in the first part of the text with a green background. (2) The description of the goal instruction with a red background shows that the task is a Cool and Place task, and the goal is to cool some pan and put it in countertop. (3) The sequence of actions and observations with a magenta background shows the actions performed and the corresponding observations from the environment. The actions are in boldface after the greater than symbol, and the observations are in regular text below each action.

Table 1: Success rate of different open-source language models with ReAct (Yao et al., 2023) applied to the ALFWorld environment (Shridhar et al., 2020) in a 12-hour run for each model

| Developer | Name of the language model | Success rate (%) | Number of success tasks | number of tasks |
|---|---|---|---|---|
| Google | gemma-2-9b | 40 | 25 | 62 |
| Google | gemma-2-9b-it | 62 | 37 | 59 |
| Mistral AI | Mistral-7B-v0.3 | 28 | 16 | 57 |
| Mistral AI | Mistral-7B-Instruct-v0.3 | 25 | 14 | 54 |
| LLaMA | Llama-2-7b-hf | 8 | 4 | 50 |
| Microsoft | Phi-3-medium-128k-instruct | 41 | 12 | 29 |
| DeepSeek | deepseek-llm-7b-base | 14 | 15 | 103 |
| Hugging Face H4 | zephyr-7b-alpha | 15 | 8 | 52 |

and place the "handtowel" in the "countertop" until the maximum step limit is reached. Third, some LLMs will misinterpret the order of sub-goals. For example, when the goal is to "examine the cd with the desklamp", the gemma-2-9b-it model erroneously attempts to retrieve a "desklamp" first. After failing to obtain the "desklamp", the LLM searches for the "cd" but only revisits previously searched locations instead of exploring new ones.

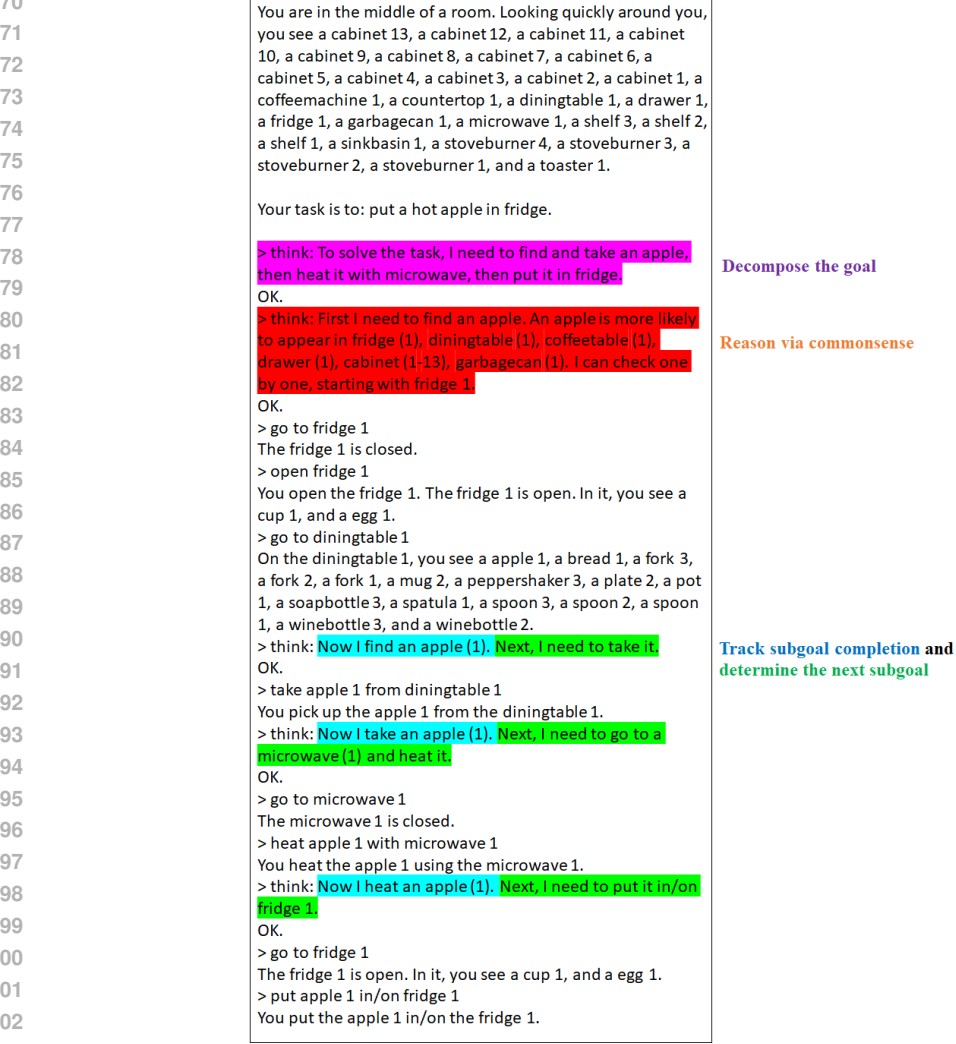

Figure 3: An exemplar in ReAct (Yao et al., 2023) for a Heat and Place task in the ALFWorld environment (Shridhar et al., 2020). The text in the black box is composed of the description from the ALFWorld environment (Shridhar et al., 2020) and the thoughts annotated by ReAct (Yao et al., 2023). The thought that decomposes the goal is shown with a magenta background. The thought that uses commonsense reasoning to find an object and determine what to do with it is shown with a red background. The thoughts that track subgoal completion are shown with a cyan background. The thoughts that determine the next subgoal are shown with a green background.

## 4.2 EXPERIMENTAL RESULT OF SALA

In Reflexion (Shinn et al., 2023), a status text indicating whether a task is successful or not, along with a reflection text that guides the next trial toward success, are concatenated after the ReAct exemplars. Reflexion (Shinn et al., 2023) used two exemplars, as shown in Figure 4 in Appendix A.1, to guide the LLM in generating the reflection.

Unlike Reflexion (Shinn et al., 2023), which uses one LLM to generate thoughts and actions and another LLM to generate reflections, SALA uses a single LLM to generate thoughts, actions, and self-adaptations by concatenating the two Reflexion exemplars (Shinn et al., 2023) after the two Re-Act exemplars (Yao et al., 2023) for each task. This approach allows the adaptation to be generated after the LLM receives the string "STATUS: FAIL". We experiment with this single-LLM setup using the exemplars from ReAct (Yao et al., 2023) and Reflexion (Shinn et al., 2023) as depicted in Figure 5 in Appendix A.1.

Table 2: Number of steps for the ReAct-based (Yao et al., 2023) agent and the proposed SALA to complete tasks in the ALFWorld environment (Shridhar et al., 2020)

| Task number | 1 | 2 | 3 | 4 | 5 | 6 | 7 |
|---|---|---|---|---|---|---|---|
| ReAct (Yao et al., 2023) | fail | fail | 14 | 10 | 12 | 19 | fail |
| SALA | 13 | 12 | fail | 10 | 17 | 87 | fail |

| Task number | 8 | 9 | 10 | 11 | 12 | 13 | 14 |
|---|---|---|---|---|---|---|---|
| ReAct (Yao et al., 2023) | 23 | fail | 10 | 19 | fail | 16 | 15 |
| SALA | 19 | fail | 10 | 17 | 67 | fail | 10 |

Initially, the input to the LLM is a prompt with the instruction, exemplars, and the description of the environment and the task to be completed. The output of the LLM will be the input prompt concatenated with a series of thoughts, actions, and observations. We extract the first thought or action from the output by extracting the line immediately following the input prompt content. The line immediately after the greater than symbol ($>$) is used as input to the environment. If the input to the environment contains the string "think:", we overwrite the output with "OK.", following the process described in (Yao et al., 2023; Shinn et al., 2023).

If the task is completed, the environment will return a variable with a value of 1, and the process is finished. Otherwise, we concatenate the extracted thought or action with the observation and append this new text to the input prompt to form the next input prompt. After 50 steps, if the task remains unfinished, we reinitialize the ALFWorld environment (Shridhar et al., 2020) and append "STATUS: FAIL" and "New plan:" after the last observation. This updated text becomes the next input prompt to the LLM. The output of the LLM will include the adaptation text following "New plan:", which is extracted and concatenated above the initial prompt, $s_0^{ep}$. This new prompt is then used as the next input to the LLM to obtain corrected thoughts and actions. We set a maximum of 9 trails; if $ep \geq 10$, we terminate the trial and proceed to the next task. The results of this experiment, conducted for 14 different tasks using gemma-2-9b-it, are shown in Table 2.

Among the fourteen tasks, ten were completed successfully, while four were not finished after reaching 10 adaptation steps. Of the ten successful tasks, eight were completed without reaching the adaptation step. Failures in tasks 7 and 9 for both methods were due to issues within the ALFWorld environment (Shridhar et al., 2020). Although both agents successfully finished the tasks for tasks 7 and 9, the environment failed to indicate the task were completed. Consequently, tasks 7 and 9 are excluded from the evaluation of different methods. Excluding these tasks, the proposed SALA achieves a success rate of approximately 83%, outperforming the ReAct-based agent, which achieved a success rate of 67%.

In one of the failed tasks (task 3) from SALA, the only adaptation obtained was: "I was stuck in a loop in which I continually looked for a lettuce in the fridge. I should have looked for a lettuce in the fridge, then taken it, then put it in the countertop. I will try to execute a different action if I am stuck in a loop again." for every trial. However, the SALA agent did not attempt different actions in subsequent trials and continued to examine the fridge. The actual issue in task 3 is that the fridge does not contain lettuce, but the SALA agent incorrectly assumes it is present and persistently attempts to retrieve it. A similar issue is also found in task number 13 which has the adaptation: "I was stuck in a loop in which I continually examined fridge 1 instead of using the fridge to cool the lettuce. I should have taken the lettuce from the fridge and then put it on the countertop. I will try to execute a different action if I am stuck in a loop again." The SALA agent did not attempt different actions in subsequent trials and continued to examine the fridge.

In task 6, the task could not be completed in the first trial but was successfully completed in the second trial with 38 steps. The goal of this task is to "find two pillow and put them in sofa". In the first trial, the agent failed to complete the task because it attempted to pick up 'pillow 2' from 'sofa 1'; however, 'sofa 1' only contained 'pillow 1'. Afterward, the agent continued trying to put 'pillow 2' on 'sofa 1,' but it failed because it did not have the pillow. Subsequently, the LLM did not output any text until the end of the trial. After this, the following adaptation was generated and appended to the input prompt for the second trial: "I was stuck in a loop in which I continually looked for the second pillow in sofa 1. I should have looked for the second pillow in armchair 1, sidetable 1, and cabinet 1-4. I will try to execute a different action if I am stuck in a loop again.". In the second trial,

the agent found that 'pillow 2' was on 'armchair 1,' picked it up, placed it on the sofa, and completed the task. The trajectories of the two trials in task number 6 are shown in Figure 6 in Appendix A.1.

In task 12, the task could not be completed in the first trial but was successfully completed in the second trial within 18 steps. The goal of this task is to "put a cool tomato in microwave". In the first trial, the agent failed to complete the task because it attempted to take a 'tomato' from 'fridge 1'; however, 'fridge 1' doesn't contained any tomato. Afterward, the agent continued trying to take a 'tomato' from 'fridge 1' until the end of the trial. After this, the following adaptation was generated and appended to the input prompt for the second trial: "I was stuck in a loop in which I continually looked for a tomato in the fridge. I should have looked for a tomato in a different environment. I will try to look for a tomato in a different environment in the next trial.". In the second trial, the agent found that 'tomato 1' was on 'countertop 1,' picked it up, cooled it with the fridge, placed it in the microwave, and completed the task. The trajectories of the two trials in task number 12 are shown in Figure 7 in Appendix A.1.

## 5 Conclusion

We present a novel in-context learning algorithm designed for a single language model to complete tasks in a text-based game by correcting its previous failures. This approach, by introducing self-adaptation reduces the number of models used in previous work (Shinn et al., 2023) from two LLMs to one LLM, gaining in autonomy. Our findings indicate that the gemma-2-9b-it model achieves the highest success rate of 62% for completing tasks in the ALFWorld environment (Shridhar et al., 2020) using ReAct prompting (Yao et al., 2023), compared to other selected open-source language models. In the twelve selected tasks from the ALFWorld environment (Shridhar et al., 2020), the proposed SALA, utilizing the gemma-2-9b-it model, achieved a success rate of 83%, outperforming the ReAct-based agent (Yao et al., 2023) with the same model, which attained a 67% success rate. In addition, we show that using SALA with gemma-2-9b-it, two of the six tasks that could not be completed in one trial can be completed in the second attempt by appending the adaptation from the previous trial. Future work will involve further experimentation with different in-context learning algorithms to complete decision-making tasks using a single language model without Reflexion exemplars for reducing the number of tokens in the input prompt.

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

# A  APPENDIX

## A.1  FIGURES

In Figure 6, the text in each black box represents one trajectory in the ALFWorld environment (Shridhar et al., 2020), as detailed in the experiment in Section 4.2. In each black box, the text at the top includes the description of available receptacles and the goal instruction, as mentioned in Fig. 2. Below the goal instruction, the text following the greater-than symbol ('>') represents the thoughts or actions generated by the gemma-2-9b-it LLM. Text beginning with '> think:' indicates a thought, while all other text beginning with without 'think' represents actions. The text beneath each thought or action is the observation from the ALFWorld environment (Shridhar et al., 2020). The trajectory of the first trial in task number 6 is shown in the left black box. The LLM agent failed to complete the task because it couldn't find the second pillow, incorrectly believing it had finished the task, and then continued generating empty strings. The adaptation the LLM agent generated is highlighted with a cyan background. In the second trial, shown in the right black box, the adaptation is appended above the description of the available receptacles. The agent successfully found the second pillow in the second trial and completed the task.

In Figure 7, the text in each black box represents one trajectory in the ALFWorld environment (Shridhar et al., 2020), as detailed in the experiment in Section 4.2. In each black box, the text at the top includes the description of available receptacles and the goal instruction, as mentioned in Fig. 2. Below the goal instruction, the text following the greater-than symbol ('>') represents the thoughts or actions generated by the gemma-2-9b-it LLM. Text beginning with '> think:' indicates a thought, while all other text beginning with without 'think' represents actions. The text beneath each thought or action is the observation from the ALFWorld environment (Shridhar et al., 2020). The trajectory of the first trial in task number 6 is shown in the left black box. The LLM agent failed to complete the task because it couldn't find a tomato, and then continued checking the same receptacle. The adaptation the LLM agent generated is highlighted with a cyan background. In the

You are in the middle of a room. Looking quickly around you, you see a bed 1, a desk 2, a desk 1, a drawer 6, a drawer 5, a drawer 4, a drawer 3, a drawer 2, a drawer 1, a garbagecan 1, a laundryhamper 1, a safe 1, a shelf 6, a shelf 5, a shelf 4, a shelf 3, a shelf 2, and a shelf 1.
Your task is to: look at bowl under the desklamp.
> think: To solve the task, I need to find and take a bowl, then find and use a desklamp.
OK.
> think: First I need to find a bowl. A bowl is more likely to appear in drawer (1-6), desk (1-2), shelf (1-6), garbagecan (1), laundryhamper (1). I can check one by one, starting with drawer 1.
OK.
> go to drawer 1
The drawer 1 is closed.
> open drawer 1
You open the drawer 1. The drawer 1 is open. In it, you see a cellphone 2.
> go to drawer 2
On the drawer 2, you see a keychain 1.
> go to drawer 3
Nothing happens.
> go to drawer 4
The drawer 4 is closed.
> open drawer 4
You open the drawer 4. The drawer 4 is open. In it, you see nothing.
> go to drawer 5
On the drawer 5, you see nothing.
> go to drawer 6
The drawer 6 is closed.
> open drawer 6
You open the drawer 6. The drawer 6 is open. In it, you see a book 2, and a pen 2.
> go to desk 1
On the desk 1, you see a alarmclock 1, a desklamp 1, a mug 1, and a pencil 1.
> think: Now I find a desklamp (1). Next, I need to use it.
OK.
> use desklamp 1
You turn on the desklamp 1.
>
Nothing happens.
> go to desk 2
On the desk 2, you see a bowl 2, a bowl 1, a cd 2, a cd 1, and a creditcard 3.
> think: Now I find a bowl (2). Next, I need to take it.
OK.
> take bowl 2 from desk 2
You pick up the bowl 2 from the desk 2.
>
Nothing happens.
>
Nothing happens.
STATUS: FAIL
New plan: In this environment, my plan was to find a bowl then find and use a desklamp. However, the task says to look at bowl under the desklamp. I should have looked for the desklamp then looked for the bowl. I noticed that the desklamp was found on desk 1. In the next trial, I will go to desk 1, find the lamp, then look for the bowl under the desklamp.

You are in the middle of a room. Looking quickly around you, you see a cabinet 10, a cabinet 9, a cabinet 8, a cabinet 7, a cabinet 6, a cabinet 5, a cabinet 4, a cabinet 3, a cabinet 2, a cabinet 1, a coffeemachine 1, a countertop 2, a countertop 1, a diningtable 2, a diningtable 1, a drawer 3, a drawer 2, a drawer 1, a fridge 1, a garbagecan 1, a microwave 1, a sinkbasin 1, a stoveburner 4, a stoveburner 3, a stoveburner 2, a stoveburner 1, and a toaster 1.
Your task is to: heat some mug and put it in coffeemachine.
> think: To solve the task, I need to find and take a mug, then heat it with stoveburner, then put it in coffeemachine.
Ok.
> look
You are in the middle of a room. Looking quickly around you, you see nothing.
> think: First I need to find a mug. A mug is more likely to appear in countertop (1-2), diningtable (1-2), drawer (1-3), fridge (1), sinkbasin (1), stoveburner (1-4), toaster (1). I can check one by one, starting with countertop 1.
Ok.
> look
You are in the middle of a room. Looking quickly around you, you see nothing.
> go to countertop 1
On the countertop 1, you see a butterknife 1, a cup 1, a fork 2, and a mug 1.
> think: Now I find a mug (1). Next, I need to take it.
Ok.
> take mug 1 from countertop 1
You pick up the mug 1 from the countertop 1.
> think: Now I take a mug (1). Next, I need to heat it with stoveburner (1-4).
Ok.
> go to stoveburner 1
On the stoveburner 1, you see a pan 2.
> examine stoveburner 1
On the stoveburner 1, you see a pan 2.
> examine stoveburner 1
On the stoveburner 1, you see a pan 2.
> examine stoveburner 1
On the stoveburner 1, you see a pan 2.
> examine stoveburner 1
On the stoveburner 1, you see a pan 2.
STATUS: FAIL
New plan: I was stuck in a loop in which I continually examined stoveburner 1 instead of heating mug 1 with stoveburner 1. I should have taken mug 1 from countertop 1, then heated it with stoveburner 1, then put it in coffeemachine 1. It did not help to execute two identical actions in a row. I will try to execute a different action if I am stuck in a loop again.

Figure 4: Two exemplars in Reflexion (Shinn et al., 2023) for the ALFWorld environment (Shridhar et al., 2020). The text in each black box comprises one exemplar from Reflexion (Shinn et al., 2023) designed to guide an LLM in generating the correct action to complete a task in the ALFWorld environment (Shridhar et al., 2020). In each black box, the text preceding the yellow-background text represents a ReAct trajectory, as shown in Fig. 3. The text next to "STATUS: " indicates whether the task is completed. If the task is completed, the yellow-background text will read "STATUS: OK". If the task is not completed, it will read "STATUS: FAIL". The reflection text is highlighted with a cyan background.

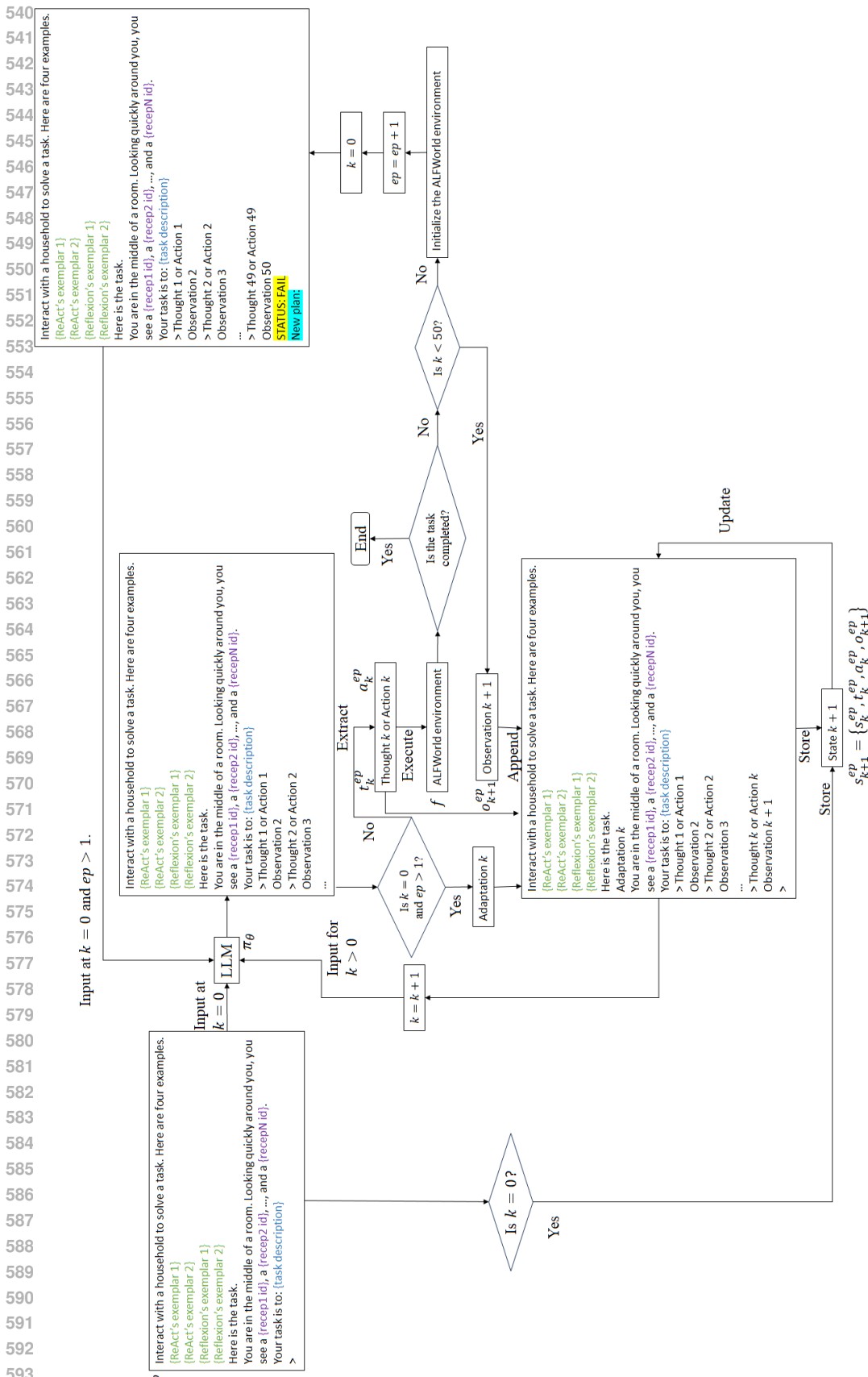

Figure 5: The process of using a single LLM to generate cycles of thoughts, actions, and self-adaptations by fusing ReAct's (Yao et al., 2023) and Reflexion's exemplars (Shinn et al., 2023) for solving tasks in the ALFWorld environment (Shridhar et al., 2020).

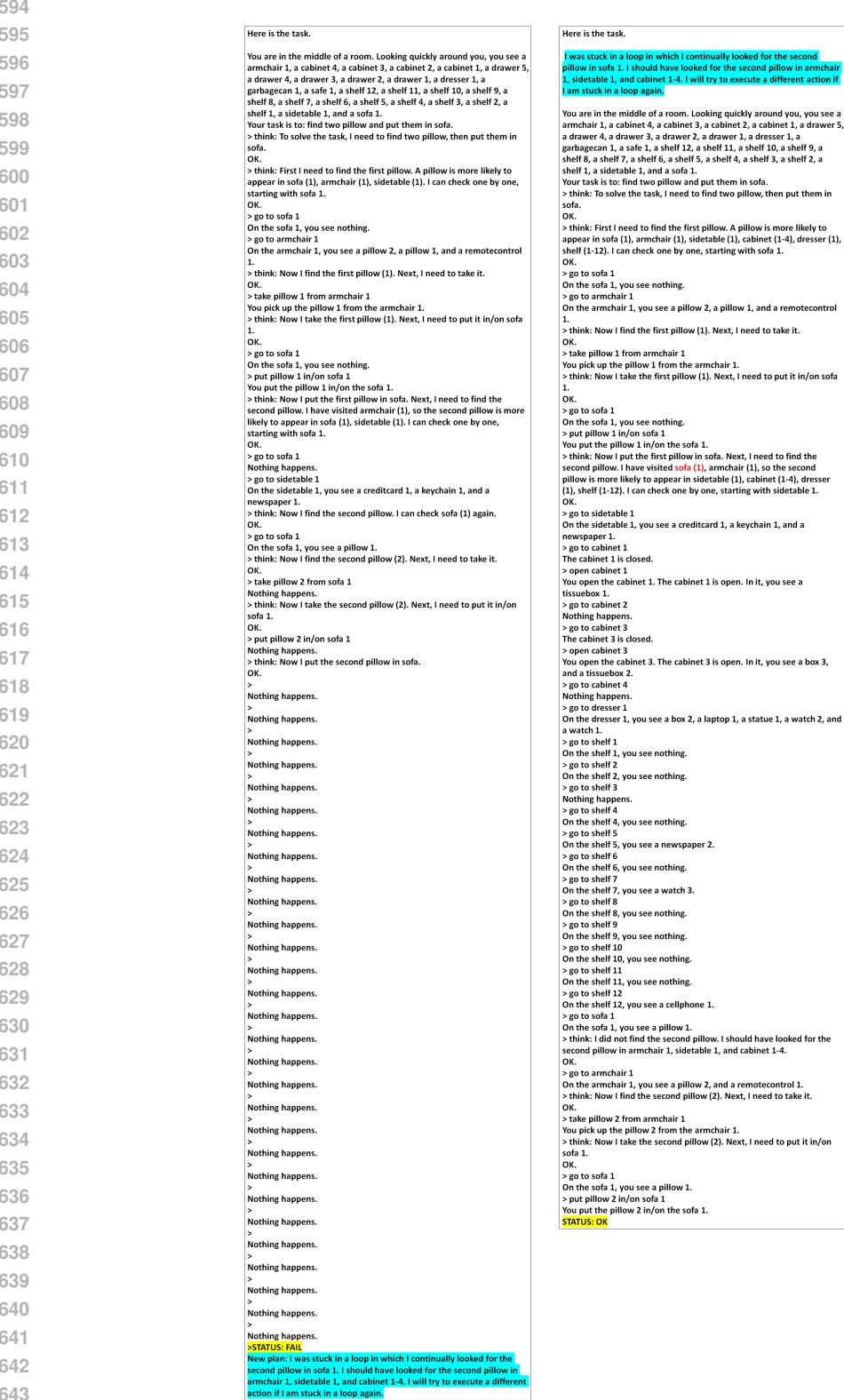

Figure 6: Two trajectories for task 6 generated by SALA. The left trajectory represents a failure, where the agent incorrectly assumes task completion and ceases actions. The right trajectory includes an adaptation (highlighted in cyan), leading to successful task completion.

second trial, shown in the right black box, the adaptation is appended above the description of the available receptacles. The agent successfully found a tomato in the second trial and completed the task.

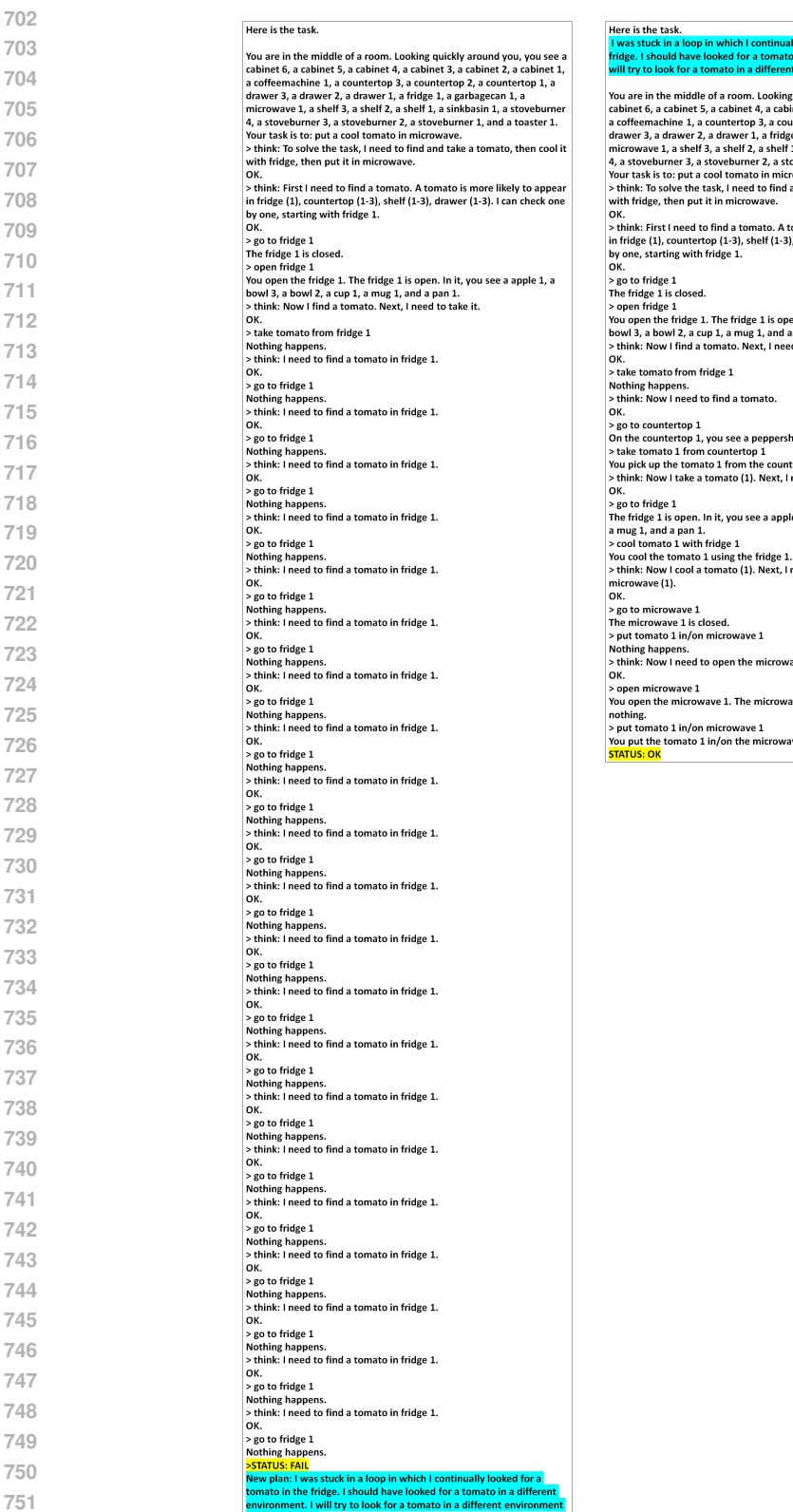

Figure 7: Two trajectories for task 12 generated by SALA. The left trajectory shows a failure, where the agent repeats the same action with no progress until the maximum step is reached. The right trajectory includes an adaptation (in cyan), leading to successful task completion.

