# OpenReview forum: "Towards Autonomous Agents: Adaptive-planning, Reasoning, and Acting in Language Models"
_ICLR.cc/2025/Conference — ICLR 2025 Conference Withdrawn Submission_

### Official Review · Reviewer_FG9y · 2024-10-25

**Soundness:** 1
**Presentation:** 2
**Contribution:** 1
**Rating:** 1
**Confidence:** 4

**Summary:**

The paper proposes an architecture towards autonomous agent (SALA), which use one large language model (LLM) to generate thoughts, actions, and self-adaptations by concatenating two Reflexion (Shinn et al., 2023) exemplars and two ReAct (Yao et al., 2023) exemplars for each task. They show that this approach is more effective than ReAct on the twelve selected tasks from ALFWorld environment and it achieves a success rate of 83%. They also find that the self-adaptation is particularly important for decision making in the next trial.

**Strengths:**

**Originality**: The paper modifies the Reflexion that uses two large language models (LLMs) to use only one LLM to generate thoughts, actions, and self-adaptations, which may use less memory. However, there are problems in the originality, which will be further discussed in the Weakness Section.

**Quality**: (Experiments are not sufficient, which will be detailed in Weaknesses Section)

**Clarity**: The paper is easy to follow. Illustrative figures are helpful for understanding the paper.

**Significance**: (The proposed method SALA lacks significance, which will be detailed in Weaknesses Section)

**Weaknesses:**

**Lack of Novelty**
- The proposed method SALA use one model for action generation and self-adaptation, but this lacks novelty in the methodology because Reflexion pipeline is sequential and one can use one LLM for different tasks with different prompts.
- Minor modification from the original method Reflexion is not considered to be novel if there's no superior results to the original method. The paper does not compare the method SALA to Reflexion on either accuracy or efficiency.

**Lack of Experiments**
- The experiments only compare the proposed method SALA with ReAct, which is less capable than Reflexion (the method augmenting the ReAct with self-reflection). Since SALA applies minor modification from the Reflexion, authors should add experiments with the Reflexion.
- There are only text-based game ALFWorld in the experiments, which is not enough to demonstrate the advantages of the proposed method. Authors should consider adding other datasets used in Reflexion like HotpotQA [1] for reasoning, and MBPP [2], HumanEval [2], LeetcodeHardGym [3] for programming.
- The paper only have results on the open-sourced language model gemma-2-9b-it, which are not enough. Authors should evaluate at least 3 open-sourced models with different model size (e.g. 30B, 65B). Additionally, it is recommend to add API models like GPT-4o since they are generally more capable than those open-sourced models which may lead to different model behaviors with different methods. At the end, authors should aggregate the results and also average them to provide a more comprehensive evaluation.

*References*

[1] Yang, Zhilin, et al. "HotpotQA: A dataset for diverse, explainable multi-hop question answering." arXiv preprint arXiv:1809.09600 (2018).

[2] Austin, Jacob, et al. "Program synthesis with large language models." arXiv preprint arXiv:2108.07732 (2021).

[3] Chen, Mark, et al. "Evaluating large language models trained on code." arXiv preprint arXiv:2107.03374 (2021).

[4] Shinn, Noah, et al. "Reflexion: Language agents with verbal reinforcement learning." Advances in Neural Information Processing Systems 36 (2024).

**Questions:**

**Questions**
- Reflexion uses three models: an Actor to generate text and actions, an Evaluator to score the outputs from the Actor and a Self-Reflection model to generate verbal reinforcement cues for self-improvement. Why the paper states that Reflexion uses only two models? The paper needs more clarifications on this discrepancy.

- What is the main motivations for modifying the Reflexion to use a single LLM to generate thought, action, and self-adaptation? Why this minor modification would differentiate this work from Reflexion? Authors should clarify the motivations.

**Suggestions**
- Authors should add a column of average success rate and average steps in the table for easier comparison (line 327).

---

### Official Review · Reviewer_Gtuc · 2024-11-02

**Soundness:** 1
**Presentation:** 1
**Contribution:** 1
**Rating:** 3
**Confidence:** 4

**Summary:**

This paper describes a new in-context learning algorithm for LLM to repeatedly attempt to solve tasks.

**Strengths:**

Overall, I don’t think this is paper is ready to be considered at ICLR yet for the reasons listed in the weakness section below.

**Weaknesses:**

* The writing of this paper is unclear. There is no proper discussion on related work. Some paragraphs in the results section, around line 370 and line 380, are completely repetitive without providing much insight.
* The proposed method seems to be minor incremental contribution on top of ReAct, with the main difference being using one model instead of two models. Large portions of the method section is repeating existing method or discussing lengthy but trivial procedures of applying LLM for solving tasks but no insight is provided about the proposed method.
* The evaluation is minimal. It is only evaluated on ALFWorld.  Additionally, it only compares against ReAct but does not consider any ablations or other baselines.
* The paper lacks content. Among its limited evaluation, large portion is devoted to evaluating ReAct, which is not proposed by this work, on different open-sourced models. This type of evaluation is useful but should not be considered as experimental contribution for ICLR.

**Questions:**

This paper would require quite significant modifications to be considered for publication IMHO.

---

### Official Review · Reviewer_J3GJ · 2024-11-04

**Soundness:** 2
**Presentation:** 2
**Contribution:** 2
**Rating:** 3
**Confidence:** 5

**Summary:**

The paper tackles the problem of planning and reasoning in a text-world environment (ALFWorld). The paper proposes to use a single large language model to output and correct its plans guided by mistakes in its generation (failures in task completion in the environment) by iteratively prompting the same model to adaptively re-plan. The paper highlights that iterative correction with an LLM allows their Agent to complete more tasks than a single round of interaction.

**Strengths:**

The paper is clearly written.

**Weaknesses:**

- The sole difference between Reflexion (Shinn et al., 2023), and the proposed approach seems to be the use of a single vs two models to provide feedback. With both the reflection and action model being the same pre-trained model, it is unclear whether this distinction is even a significant one--both the methods require the same number of inference steps with the same model. If there are other differences, please indicate them in the paper and additionally include a quantitative comparison to Reflexion as a baseline.
- Missing implementation and dataset details: There is little clarity on the metrics being reported here.
  - How is the accuracy computed?
  - Does Table 2 report the average number of steps across data points in each task type?
  - What is the size of the test set?
  - Missing details around the prompt provided to the model for feedback?
- Why are the standard metrics for testing task performance on ALFWorld / Alfred not reported?
- What is the task success rate of the methods across each of the task types?
- The paper states that it proposes a "new in-context learning algorithm". The description needs to be more measured. There isn't a new in-context learning algorithm, but a modified prompting strategy.

**Questions:**

Please take a look at the section on weaknesses.

---

### Official Review · Reviewer_MVzq · 2024-11-06

**Soundness:** 3
**Presentation:** 2
**Contribution:** 2
**Rating:** 3
**Confidence:** 4

**Summary:**

The paper introduces a novel in-context learning algorithm for building autonomous decision-making agents using a single language model (SALA). The approach integrates reasoning, acting, and self-adaptation mechanisms to enhance problem-solving capabilities in text-based environments, achieving higher success rates compared to existing methods like ReAct and Reflexion.

**Strengths:**

1. The proposed SALA method effectively combines reasoning, acting, and adaptation within a single language model, reducing model complexity while achieving autonomy.

2. The integration of a correction mechanism enhances decision-making capabilities, demonstrating significant improvements over prior methods, such as Reflexion. Besides, experimental results show an 83% success rate, outperforming ReAct by a substantial margin, thus demonstrating the practical significance of SALA in task completion.

3. The use of self-adaptation allows for reducing resource requirements compared to Reflexion, which relies on two LLMs, showing efficient use of model capabilities.

**Weaknesses:**

1. The introduction lacks a clear articulation of the specific limitations of existing methods that the proposed SALA intends to address, making it difficult to understand the motivation behind the approach.

2. The experimental results are poorly presented, with insufficient statistical analysis to demonstrate the robustness of the reported success rates, leaving room for questioning the significance of the results.

3. Lack adequate details on the hyperparameter settings or the training configurations, which makes it challenging to reproduce the experiments or assess the validity of the claims. The comparison with baselines is superficial, lacking a comprehensive evaluation against more recent methods that have demonstrated competitive results in similar environments.

4. The paper fails to justify the use of a single LLM for both action generation and self-adaptation in terms of scalability or performance, especially when dealing with more complex decision-making tasks.

**Questions:**

1. The experimental results lack detailed statistical significance testing. Can you provide additional quantitative analysis, such as confidence intervals or statistical tests, to verify the robustness of the reported success rates for SALA compared to other baselines?
In your experimental setup, hyperparameter details and training configurations are not clearly outlined. Could you specify these parameters, including learning rates, training duration, and the criteria used to choose these values, to ensure reproducibility?

2. The comparison with baselines is somewhat limited. Have you considered evaluating SALA against more recent, high-performing language model agents beyond those mentioned?

3. The introduction touches briefly on the limitations of existing methods but lacks specificity. Could you elaborate more explicitly on the shortcomings of prior methods and explain how each of these motivates the design choices in SALA? This would help clarify the intended contributions and situate them within the broader research context.

**Details Of Ethics Concerns:**

No.

---

### Note · Authors · 2024-11-25

I have read and agree with the venue's withdrawal policy on behalf of myself and my co-authors.